# *Staphylococcus argenteus* Infections, Brazil

Lisiane da Luz Rocha Balzan,[a] Adriana Medianeira Rossato,[a] Cezar Vinicius Wurdig Riche,[a] Vlademir Vicente Cantarelli,[a] Pedro Alves D'Azevedo,[a] Aline Valério de Lima,[b] Beatriz Rodrigues,[c] Ivan Leonardo Avelino França e Silva,[d] Cícero Armídio Gomes Dias,[a] Jorge Luiz Mello Sampaio[b,e]

[a]Universidade Federal de Ciências da Saúde de Porto Alegre, Porto Alegre, Brazil

[b]Faculdade de Ciências Farmacêuticas, Universidade de São Paulo, São Paulo, Brazil

[c]São Camilo Oncologia, São Paulo, Brazil

[d]AC Camargo Cancer Center, São Paulo, Brazil

[e]Grupo Fleury, São Paulo, Brazil

**ABSTRACT** In 2015, two new species related to the *Staphylococcus aureus* were proposed. We describe five isolates of the new species *Staphylococcus argenteus* cultured from human cases of bacteremia and skin and soft tissue infections. This is the first report of *S. argenteus*, from South America, causing community-acquired and nosocomial infections.

**KEYWORDS** *Staphylococcus argenteus*, *Staphylococcus aureus*, *Staphylococcus aureus* complex

In 2015, two new species related to the *Staphylococcus aureus* complex were proposed: *S. argenteus* and *S. schweitzeri* (1). The modification was primarily based on the considerable genetic distance at the whole-genome sequencing level. On the other hand, the new species are phenotypically similar and coagulase positive and have nearly identical rRNA 16S gene sequences compared to *S. aureus*. *S. argenteus* is a human pathogen, whereas *S. schweitzeri* has been reported to cause infections only in nonhuman primates. Several methods have been employed to identify *S. argenteus*, including detecting and sequencing specific genes. Mass spectrometry can differentiate *S. argenteus* from closely related species but requires a visual comparison of spectra (2). Some studies have retrospectively evaluated the presence of *S. argenteus* in collections of isolates previously identified as *S. aureus* in different countries. The results are variable, with studies indicating a low proportion of the new species in countries such as Belgium (3/1,903, 0.16%, unspecified infections) (3), Japan (2/201, 1%, bacteremia) (4), and Sweden (18/5,500, 0.3%, unspecified infections) (5). In contrast, the proportion of isolates identified as *S. argenteus* was high in two studies conducted in Thailand (10/246, 4.1% and 58/311, 19%, bacteremia) (6), and Laos (6/96, 6.3% skin and soft tissue infections) (7). There is a relationship between *S. argenteus* and *S. aureus* ST 2250, although other STs have been identified worldwide (8). No reports on *S. argenteus* human infections have emerged from Latin America. This study aims to describe and characterize clinical isolates of *S. argenteus* obtained from patients in Brazil. We evaluated five clinical isolates from two Brazilian states cultured from samples received from November 2018 to June 2019 initially identified by the Bruker MALDI Biotyper as *S. argenteus/S. schweitzeri*, *S. argenteus*, or *S. schweitzeri*. Three isolates were detected from inpatients from the city of São Paulo, São Paulo (SP) state. The other two were detected from inpatients from the city of Porto Alegre, Rio Grande do Sul (RS) state, in the southern part of Brazil.

Species identification was performed by partial sequencing of the gene encoding the hypothetical nonribosomal peptide synthetase (NRPS), as previously described by Zhang et al. (9). PCR products were sequenced using the same primers used for amplification. The obtained sequences were translated and compared to those from *S. argenteus*

Address correspondence to Lisiane da Luz Rocha Balzan, lisi.rocha@hotmail.com.

The authors declare no conflict of interest.

**TABLE 1** Sample, infection type, matrix-assisted laser desorption ionization–time of flight mass spectrometry identification, and sample collection date[a]

| Isolate | Patient | Collection date | City of isolation | Clinical sample | Infection type | MALDI ID | Presence of *mecA*, *lukS-lukF* genes | Susceptibility (S) or resistance (R) | |
|---|---|---|---|---|---|---|---|---|---|
| | | | | | | | | S | R |
| 01SAP | 1 | 06/19/2019 | São Paulo | Mastitis secretion | HAI | *S. argenteus/ S. schweitzeri* | Negative | FOX, CIPO, CLIN, ERY, GEN, LZD, PEN, TET, SXT, VAN | |
| 02SAP | 2 | 12/12/2018 | São Paulo | Blood | CAI | *S. argenteus* | Negative | FOX, CIPO, CLIN, ERY, GEN, LZD, TET, SXT, VAN | PEN |
| 07SAP | 2 | 12/12/2018 | São Paulo | Skin abscess | CAI | *S. argenteus* | Negative | FOX, CIPO, CLIN, ERY, GEN, LZD, PEN, TET, SXT, VAN | |
| 01POA | 3 | 05/06/2019 | Porto Alegre | Umbilical cord | HAI | *S. schweitzeri* | Negative | FOX, CIPO, CLIN, ERY, GEN, LZD, SXT, VAN | PEN, TET |
| 02POA | 4 | 05/27/2019 | Porto Alegre | Blood | HAI | *S. argenteus* | Negative | FOX, CIPO, CLIN, ERY, GEN, LZD, PEN, TET, SXT, VAN | |

[a]MALDI ID, matrix-assisted laser desorption ionization–time of flight mass spectrometry identification; HAI, Healthcare-associated infection; CAI, Community-acquired infection; FOX, cefoxitin; CIP, ciprofloxacin; CLIN, clindamycin; ERY, erythromycin; GEN, gentamicin; LZD, linezolid; PEN, penicillin; TET, tetracycline; SXT, trimethoprim-sulfamethoxazole; VAN, vancomycin.

(WP_000605287.1) and *S. schweitzeri* (VEE64761.1) type strains, available at GenBank, using the BLAST program (10). The presence of the *mecA* and *lukS-lukF* genes were evaluated by PCR as previously described by Boakes et al. Antimicrobial susceptibility testing was performed by disk diffusion, according to the European Committee on Antimicrobial Susceptibility Testing (EUCAST), for cefoxitin (FOX), ciprofloxacin (CIP), clindamycin (CLIN), erythromycin (ERY), gentamicin (GEN), linezolid (LZD), penicillin (PEN), tetracycline (TET), and trimethoprim-sulfamethoxazole (SXT). Vancomycin (VAN) susceptibility was evaluated by broth microdilution as recommended by the EUCAST (11). From November 2018 to June 2019, five clinical isolates were recovered from blood culture ($n = 2$), mastitis secretion ($n = 1$), skin abscess ($n = 1$), and umbilical cord ($n = 1$). Three isolates were recovered from health care-associated infections, according to U.S. Centers for Disease Control and Prevention criteria. Two isolates were recovered from a blood sample and from a cutaneous abscess from a patient with a community-acquired infection from São Paulo (Table 1). All isolates were positive for catalase and coagulase tests and produced nonpigmented colonies. All five Brazilian isolates had identical partial NRPS amino acid sequences (GenBank accession submission ID 2606804). Of note, the presence of this fragment in NRPS excludes *S. aureus*. The sequences had 98.0% similarity compared to the amino acid sequence from the *S. argenteus* type strain (GenBank WP_000605287.1) and differed by amino acid substitutions N1568Y and G1612E (see Fig. S1 in the supplemental material). Amino acid sequences 100% identical to those we report, including substitutions N1568Y and G1612E, have been reported from Thailand, China, Sweden, and Japan for *S. argenteus* (GenBank numbers SGW87277.1, OMH90911.1, OAF00969.1, and BBD85000). On the other hand, a low similarity (71.3%) was observed when amino acid sequences from Brazilian isolates were compared to that of the *S. schweitzeri* type strain (GenBank VEE64761.1) (Fig. 1). All isolates were susceptible to FOX, CIP, CLIN, ERY, GEN, LZD, and SXT. Two isolates were resistant to PEN, and one was also resistant to TET. Although all of the isolates we report were methicillin susceptible, a retrospective analysis performed by Giske et al. demonstrated the detection of

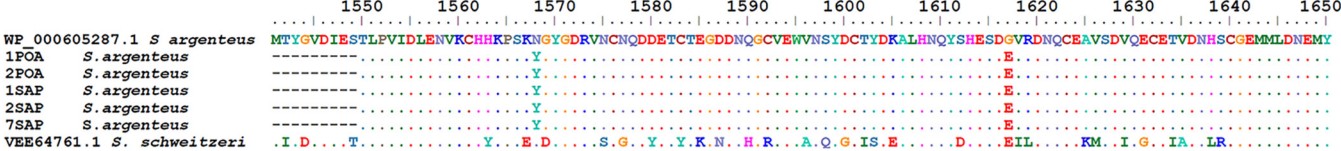

**FIG 1** Alignment of partial amino acid sequences of NRPS from Brazilian isolates and *S. argenteus* and *S. schweitzeri* type strains. Dots indicate identical amino acids. Numbers refer to NRPS amino acid numbering from *S. argenteus* type strain GenBank WP_000605287.1.

resistant strains in Europe from 2007 (12). All isolates in this study had VAN MICs of ≤1.0 mg/L. None of them had *mecA* or PVL genes.

To the best of our knowledge, this is the first report of *S. argenteus* from South America, causing community-acquired and health care-associated infections in humans. Our findings highlight the need for continued evaluation of the frequency of this species in human diseases.

## SUPPLEMENTAL MATERIAL

Supplemental material is available online only.

**SUPPLEMENTAL FILE 1**, PDF file, 0.2 MB.

## ACKNOWLEDGMENT

We thank Fleury Group for the financial support and isolates used in this study.

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
