## [Reviewer comments · Microbiology Spectrum]

Microbiology Spectrum

Staphylococcus argenteus infections, Brazil

Lisiane Rocha, Adriana Rossato, Cezar Riche, Vlademir Cantarelli, Pedro d'Azevedo, Aline V. de Lima, Beatriz Rodrigues, Ivan França e Silva, Cícero Dias, and Jorge Sampaio

Corresponding Author(s): Lisiane Rocha, Universidade Federal de Ciências da Saúde de Porto Alegre

Review Timeline:

Submission Date:	March 30, 2022
Editorial Decision:	May 24, 2022
Revision Received:	August 29, 2022
Editorial Decision:	October 25, 2022
Revision Received:	December 15, 2022
Accepted:	December 27, 2022

Editor: Rebekah Martin

Reviewer(s): The reviewers have opted to remain anonymous.

Transaction Report:

DOI: <https://doi.org/10.1128/spectrum.01179-22>

May 24, 2022

Dr. Lisiane da Luz Rocha
UFCSPA
R. Sarmiento Leite, 245 - Centro Histórico
Porto Alegre, RS 90050-170
Brazil

Re: Spectrum01179-22 (Staphylococcus argenteus infections, Brazil)

Dear Dr. Lisiane da Luz Rocha:

Thank you for submitting your manuscript to Microbiology Spectrum. Three individuals with expertise in this field reviewed the manuscript and the consensus was that modifications are necessary. When submitting the revised version of your paper, please provide (1) point-by-point responses to the issues raised by the reviewers as file type "Response to Reviewers," not in your cover letter, and (2) a PDF file that indicates the changes from the original submission (by highlighting or underlining the changes) as file type "Marked Up Manuscript - For Review Only". Please use this link to submit your revised manuscript - we strongly recommend that you submit your paper within the next 60 days or reach out to me. Detailed instructions on submitting your revised paper are below.

Link Not Available

Sincerely,

Rebekah Martin

Journals Department
Reviewer comments:

Please deposit sequenced genomes to public database and include a data availability statement.

Reviewer #1 (Comments for the Author):

No comments or suggestions

Reviewer #2 (Comments for the Author):

This is a letter to the Editor describing *S. argenteus* and *S. schweitzeri* in South America (Brazil)

Major comments:

Given emergence of pathogen in South America would urge whole genome sequencing and deposit into public repository
The 01SAP is listed as *S. argenteus*/*S. schweitzeri* - whole genome sequencing would resolve this distinction
Please explain how isolates were identified, as they were from two different states in Brazil. Was this from a central microbiology lab or central public health department?
Has there been a look back to re-identify potential cases of *S. argenteus* and *S. schweitzeri* in response to these findings?
What is the standard Micro lab work up at the two hospitals where the isolates were identified? Is MALDI-Tof standard performed on all isolates?
Would it be possible to add more clinical data for cases if appropriate IRB approvals are in place?
Line 79 describe why it infection defined as a "nosocomial infection"
Line 79-80 and on the Table it is unclear which isolates are from which patients
Please comment on isolate susceptibility patterns with relation to historically identified isolates.

Minor comments:

Add reference for BLAST
Add reference for EUCAST
Line 76 is repetitive and can be removed, as same wording provided in line 59
Line 61 correctly reference the Bruker: Bruker MALDI Biotyper®
Please add full antibiotic profiles to the Table

Reviewer #3 (Comments for the Author):

In their study, the authors characterized five isolates identified as *Staphylococcus aureus* from Brazil. They present that it is the first isolation of this species in the country, from two distinct states. All were recovered from infections, either nosocomial or community associated.
They used MALDI-TOF-MS for initial characterization, then sequenced an internal fragment of NRPS, to conclude that the strains are *Staphylococcus argenteus* and not *S. aureus* or *S. schweitzeri*.
Two amino acids mutations were found in NRPS protein sequence which are all found in *S. schweitzeri*.
Here are some concerns (mostly minor):
Reference 1 could be added at the end of the first sentence (line 42).
Reference 9 on line 67: I don't think it is the most appropriate. The authors should add this PMID (27561462) from the same team.
Line 90: the results regarding the absence of *mecA* and *pvl* genes should be added in table 1.
Table 1: Change MALDI-Tof by MALDI-TOF
Table 1: Please add the places where the isolates were recovered.
Figure: The authors should search whether the 2 mutations (N1568Y and G1612E) are found in other *S. argenteus* and then write at least a sentence on the fact that those two mutations are present in *S. schweitzeri*. This could bring some more interest to the reader.

Staff Comments:

Preparing Revision Guidelines

For complete guidelines on revision requirements, please see the journal Submission and Review Process requirements at

<https://journals.asm.org/journal/Spectrum/submission-review-process>. **Submissions of a paper that does not conform to Microbiology Spectrum guidelines will delay acceptance of your manuscript. "**

Please return the manuscript within 60 days; if you cannot complete the modification within this time period, please contact me. If you do not wish to modify the manuscript and prefer to submit it to another journal, please notify me of your decision immediately so that the manuscript may be formally withdrawn from consideration by Microbiology Spectrum.

Response to Reviewers

Please deposit sequenced genomes to public database and include a data availability statement.

Response to the reviewer: We thank the reviewer for the comment. In this work, we did not sequence the genomes of the isolates. Instead, we sequenced the PCR products of the gene encoding the non-ribosomal peptide synthetase as described by Zhang et al. All nucleotide sequences from our strains were identical. Consequently, only the nucleotide sequence from the first isolate (2SAP) was deposited at GenBank accession submission ID 2606804.

Reviewer #1

No comments or suggestions.

Reviewer #2

Given emergence of pathogen in South America would urge whole genome sequencing and deposit into public repository. The 01SAP is listed as *S. argenteus*/*S. schweitzeri* - whole genome sequencing would resolve this distinction.

Response to the reviewer: We thank the reviewer for the comment. We agree that whole genome sequencing would add more information, but it is beyond the scope of this manuscript. Isolate 01SAP is listed as *S. argenteus*/*S. schweitzeri* in the Table to indicate the result reported by the Microflex system. The figure with amino acid sequences alignment shows that those from the isolates we describe differ from the sequence of *S. schweitzeri* type strain by 28 amino acids but differ only by two amino acids when compared to that pertaining to the *S. argenteus* type strain.

Please explain how isolates were identified, as they were from two different states in Brazil. Was this from a central microbiology lab or central public health department?

Response to the reviewer: Isolates were detected in two laboratories of a national private laboratory network. Both laboratories are equipped with Bruker's Microflex System. This system initially identified all isolates as possible *S. argenteus*. We confirmed the species identification by PCR and partial sequencing of the gene encoding the non-ribosomal peptide synthetase as described by Zhang et al.

Has there been a look back to re-identify potential cases of *S. argenteus* and *S. schweitzeri* in response to these findings?

Response to the reviewer: Although we agree that it would be an excellent approach to define the frequency of misidentification of *S. argenteus* as *S. aureus*, we are doing this prospectively because we did not save all *S. aureus* isolates routinely.

What is the standard Micro lab work up at the two hospitals where the isolates were identified? Is MALDI-ToF standard performed on all isolates?

Response to the reviewer: Yes. The Bruker's Microflex system routinely identifies all bacterial isolates in both hospital labs. We started to see *S. argenteus* reports after the manufacturer implemented a database update.

Would it be possible to add more clinical data for cases if appropriate IRB approvals are in place?

Response to the reviewer: Thank you for the question. Although we agree that including more clinical information would be of interest, we were only allowed to include the information provided in the Table.

Line 79 describe why it infection defined as a "nosocomial infection"

Response to the reviewer: We updated the term to healthcare-associated infections. Infection control services from both hospitals follow CDC's definitions for HAI.

Line 79-80 and on the Table it is unclear which isolates are from which patients

Response to the reviewer: Thank you for the comment. We have now modified the Table for clarity.

Please comment on isolate susceptibility patterns with relation to historically identified isolates.

Response to the reviewer: Thank you for the suggestion. We included the following sentence: "Although all isolates we report were methicillin-susceptible, a retrospective analysis performed by Giske et al. demonstrated the detection of resistant strains in Europe from 2007."

Minor comments:

Add reference for BLAST

Answer to the reviewer: Not possible due to journal restrictions

Add reference for EUCAST

Answer to the reviewer: Not possible due to journal restrictions

Line 76 is repetitive and can be removed, as same wording provided in line 59

Answer to the reviewer: Thank you for the suggestion. We removed the sentence on line 76, as suggested.

Line 61 correctly reference the Bruker: Bruker MALDI Biotyper®

Answer to the reviewer: Thank you for the suggestion. It was corrected, as suggested.

Please add full antibiotic profiles to the Table

Answer to the reviewer: we included this information in the Table as suggested.

Reviewer #3

Here are some concerns (mostly minor):

Reference 1 could be added at the end of the first sentence (line 42). –

Answer to the reviewer: Thank you for the suggestion. It was corrected as suggested.

Reference 9 on line 67: I don't think it is the most appropriate. The authors should add this PMID (27561462) from the same team.

Answer to the reviewer: Thank you for the suggestion. It was corrected as suggested.

Line 90: the results regarding the absence of *mecA* and *pvl* genes should be added in table 1.

Answer to the reviewer: Thank you for the suggestion. We included this information in the Table.

Table 1: Change MALDI-Tof by MALDI-TOF

Answer to the reviewer: Thank you for the suggestion. It was corrected as suggested.

Table 1: Please add the places where the isolates were recovered.

Answer to the reviewer: Thank you for the suggestion. We included this information in the Table.

Figure: The authors should search whether the 2 mutations (N1568Y and G1612E) are found in other *S. argenteus* and then write at least a sentence on the fact that those two mutations are present in *S. schweitzeri*. This could bring some more interest to the reader.

Answer to the reviewer: Thank you for the suggestion. We included the following sentence: "Amino acid sequences 100% identical to those we report, including substitutions N1568Y and G1612E have been reported from Thailand, China, Sweden and Japan for *S. argenteus* (GenBank deposits SGW87277.1; OMH90911.1; OAF00969.1 and BBD85000)."

October 25, 2022

Dr. Lisiane da Luz Rocha
UFCSPA
R. Sarmiento Leite, 245 - Centro Histórico
Porto Alegre, RS 90050-170
Brazil

Re: Spectrum01179-22R1 (Staphylococcus argenteus infections, Brazil)

Dear Dr. Lisiane da Luz Rocha:

Please insert the URL for BLAST as well as a citation for the EUCAST method used, as discussed.

Thank you for submitting your manuscript to Microbiology Spectrum. As you will see your paper is very close to acceptance. Please modify the manuscript along the lines I have recommended. As these revisions are quite minor, I expect that you should be able to turn in the revised paper in less than 30 days, if not sooner. If your manuscript was reviewed, you will find the reviewers' comments below.

When submitting the revised version of your paper, please provide (1) point-by-point responses to the issues raised by the reviewers as file type "Response to Reviewers," not in your cover letter, and (2) a PDF file that indicates the changes from the original submission (by highlighting or underlining the changes) as file type "Marked Up Manuscript - For Review Only". Please use this link to submit your revised manuscript. Detailed instructions on submitting your revised paper are below.

Link Not Available

Sincerely,

S. Wesley Long

Reviewer comments:

Preparing Revision Guidelines

Please return the manuscript within 60 days; if you cannot complete the modification within this time period, please contact me. If you do not wish to modify the manuscript and prefer to submit it to another journal, please notify me of your decision immediately so that the manuscript may be formally withdrawn from consideration by Microbiology Spectrum.

December 27, 2022

Dr. Lisiane da Luz Rocha
Universidade Federal de Ciencias da Saude de Porto Alegre
R. Sarmiento Leite, 245 - Centro Histórico
Porto Alegre, RS 90050-170
Brazil

Re: Spectrum01179-22R2 (Staphylococcus argenteus infections, Brazil)

Dear Dr. Lisiane da Luz Rocha:

Your manuscript has been accepted, and I am forwarding it to the ASM Journals Department for publication. You will be notified when your proofs are ready to be viewed.

Sincerely,

S. Wesley Long
Editor, Microbiology Spectrum
